# The Relationship between the Plasma Concentration of Electrolytes and Intensity of Sleep Bruxism and Blood Pressure Variability among Sleep Bruxers

**DOI:** 10.3390/biomedicines10112804

**Published:** 2022-11-04

**Authors:** Justyna Kanclerska, Mieszko Wieckiewicz, Anna Szymanska-Chabowska, Rafal Poreba, Pawel Gac, Anna Wojakowska, Grzegorz Mazur, Helena Martynowicz

**Affiliations:** 1Department of Internal Medicine, Occupational Diseases, Hypertension and Clinical Oncology, Wroclaw Medical University, 213 Borowska St., 50-556 Wroclaw, Poland; 2Department of Experimental Dentistry, Wroclaw Medical University, 26 Krakowska St., 50-425 Wroclaw, Poland; 3Department of Population Health, Division of Environmental Health and Occupational Medicine, Wroclaw, Medical University, 7 Mikulicza-Radeckiego St., 50-345 Wroclaw, Poland

**Keywords:** sleep bruxism, hypertension, blood pressure, polysomnography, sodium, magnesium, potassium, calcium

## Abstract

Plasma sodium plays a major role in regulating blood pressure (BP). An augmented variability of BP is considered a risk factor for the development of arterial hypertension, which is prevalent among patients with suspected sleep bruxism (SB). The aims of this study were to assess the plasma concentration of electrolytes and their effect on the intensity of SB and BP variability among sleep bruxers. A total of 51 patients were enrolled in this prospective, observational study. A single full-night polysomnographic examination was conducted in the Wroclaw Medical University Sleep Laboratory, Poland, and based on the guidelines of the American Academy of Sleep Medicine, the results were analyzed. The monitoring of ambulatory BP was performed the next day, and the plasma levels of sodium, potassium, magnesium, and calcium were measured. The mean age of the studied group was 33.9 ± 11.2 years, and the mean bruxism episode index (BEI) was 4.94 ± 3.53. The study revealed statistically significant differences in the plasma concentrations of sodium in the SB group. A decreased sodium concentration was observed in the group of mild bruxers (2 ≤ BEI < 4) (139.7 ± 1.4 vs. 142.8 ± 3.2, *p* = 0.002) and severe bruxers (BEI ≥ 4) (140.5 ± 2.0 vs. 142.8 ± 3.2, *p* = 0.016) compared to nonbruxers (BEI < 2). A statistically significant positive linear correlation was found between plasma sodium concentration and daytime systolic BP variability (r = 0.32, *p* < 0.05) as well as between plasma sodium concentration and nighttime diastolic BP variability (r = 0.31, *p* < 0.05). The preliminary results suggest the probable relationship between the lower plasma concentration of sodium and increased intensity of SB and BP variability among suspected sleep bruxers.

## 1. Introduction

The plasma level of sodium plays a major role in regulating blood pressure (BP). Evidence suggests that even small increases in plasma sodium concentration may directly affect BP [1]. This can be attributed to the tendency for an increase in extracellular fluid volume which may have a pressor effect itself. A rise in sodium plasma may also induce pressor effects on the brain and on the renin-angiotensin-aldosterone system [2]. Furthermore, the intake of a high-salt diet intake has a direct harmful effect on the cardiovascular system as it indeed could increase the mass of the ventricular wall, which is an interesting fact to note and has been proven [3]. It also has an effect on thickening and narrowing resistance arteries, as well as stiffening conduit arteries—independent of, and additive to, the effect of salt on BP [4]. Moreover, increased intake of dietary sodium may increase the number of strokes, raising the severity of the cardiac failure and stimulating the tendency of platelets to aggregate [5]. The increase in BP due to the high plasma level of sodium can also be explained by the disability of the kidneys to excrete salt and the compensatory response required to restore the balance of sodium [6]. It was previously shown that among patients with insomnia, lower levels of sodium are associated with poor clinical outcomes [7]. In addition, there is a possibility of the activation of the autonomic nervous system in patients with insomnia, which could also be associated with both lower levels of serum sodium and increased risk of mortality. Sleep bruxism (SB) is considered an interdisciplinary behavior because of its central origin and the association between SB and cardiovascular diseases [8]. It often requires the involvement of a few specialists such as a cardiologist, specialist of dental diseases, as well as a specialist of internal diseases. Headache, masticatory muscle pain, and tooth damage are frequent symptoms of SB; however, the relationship between SB and temporomandibular disorders remains controversial [9]. Recent studies suggest that SB is secondary to sleep-related microarousals [10]. The etiology of SB is sympathetic system dominance [11,12]. Moreover, SB is associated with BP fluctuations during sleep. Arousals and body movements observed with SB can impact the magnitude of BP. In a study by Michałek-Zrąbkowska et al., an association between SB intensity and BP variability was found. The variability of systolic blood pressure (SBP) at nighttime was significantly increased in severe sleep bruxers; however, mean arterial BP was not increased compared to nonbruxers [13]. Arterial hypertension is linked to many sleep disturbances such as short sleep duration [14], insomnia [15], restless leg syndrome [16], and obstructive sleep apnea [17]. Thus, patients with SB may more likely to develop arterial hypertension due to coexisting pathophysiological factors [18]. Also, there is a strong connection between SB and snoring [19] that could also affect the fluctuation of blood pressure throughout the night and could be a probable risk factor for the development of OSA.

This study aimed to assess the plasma concentration of electrolytes and their effect on the intensity of SB and BP variability among suspected sleep bruxers. The null hypothesis that was set before the study started was that plasma sodium concentration has an association with BP in SB. The primary aim was to evaluate the association of electrolytes plasma concentration level on the fluctuations of blood pressure among sleep bruxers.

## 2. Materials and Methods

In this prospective, observational study, a total of 51 patients were recruited by qualified dentists in the Clinic of Prosthetic Dentistry, Department of Prosthetic Dentistry, Wroclaw Medical University, Poland. The main inclusion criterion for the patients to be included in the study was the diagnosis of suspected SB based on positive clinical inspection [20]. Examinated patients were divided into three groups among whom we could specify the mild bruxism group (2 ≤ BEI < 4, *n* = 19), the severe bruxism group (BEI ≥ 4, *n* = 24), and the control group that did not present bruxism events (BEI < 2, *n* = 8). This study was approved by the Wroclaw Medical University Bioethical Committee, Poland (no. ID KB-407/2022) and was conducted in accordance with the Declaration of Helsinki. All patients signed an informed consent form before participation.

### 2.1. Study Participants

The included patients were referred to video polysomnography in the Sleep Laboratory of the Department and Clinic of Internal Medicine, Occupational Diseases, Hypertension and Clinical Oncology at the Wroclaw Medical University, Poland. The inclusion criteria were age above 18 years, presence of probable SB, and written consent for examination. The exclusion criteria were the presence of severe mental disorders, active malignancy, respiratory and/or cardiac insufficiency, active inflammation, hypothyreosis, or chronic kidney disease; pregnancy; the use of medication that could affect the levels of sodium (e.g., diuretics, antileptics, and selective serotonin reuptake inhibitors (SSRIs)) and neuromuscular function; and inability to undergo polysomnographic examination. While admitted to the Sleep Laboratory, blood samples were taken from patients by venipuncture, and the plasma concentrations of sodium (mmol/L), calcium (mg/dL), potassium (mmol/L), and magnesium (mg/dL) were determined at the main laboratory of the University Clinical Hospital, Wroclaw, Poland.

### 2.2. Polysomnographic Examination

Polysomnograms were assessed in 30-s epochs according to the American Academy of Sleep Medicine (AASM) 2013 standard criteria for sleep scoring using Nox-A1 (Nox Medical, Reykjavík, Iceland). The bioelectrical function of the brain was recorded by electroencephalography; eye movements were evaluated by electrooculography; muscular tension was assessed from tibial electrodes (electromyogram); airflow was recorded from the nasal pressure sensor, and chest and abdomen movements were assessed by inductive plethysmography. The body position of the patient was recorded, and bilateral masseter electromyography (EMG) was also performed. SB was determined based on EMG, and the audio and video evaluation bruxism episodes were scored according to the standards of the AASM in the following three forms: phasic, tonic, and mixed. For SB, EMG bursts should not be separated by >3 s to be considered as part of the same episode, and the EMG activity had to be at least twice the amplitude of the background EMG. The scoring and manual analysis of the collected data were performed by a qualified physician (author HM) from the Sleep Laboratory, Wroclaw Medical University, Poland, in accordance with the AASM guidelines on the Noxturnal system (Version 5.1, Nox Medical, Reykjavík, Iceland)).

### 2.3. 24-h BP Examination

The next day, after performing the polysomnographic examination, the BP of the patients was monitored for 24 h. The ambulatory blood pressure monitoring (ABPM) device (90227 Ontrak, Spacelabs Healthcare, Snoqualmie, WA, USA) was installed by a qualified technician, and the measurement was performed according to the recommendations of the European Society of Hypertension [21]. After the appropriate selection of the cuff by the technician (measuring the circumference of the upper arm), all participants were instructed about the correct position during BP measurement. The measurement time intervals were 15 and 20 min for daytime and nighttime, respectively. The daytime period was recorded as the period between the time of attachment of the device by the patient and the time when the patient went to bed, while the nighttime period was registered as the period between the time when the patient went to bed and the time when the patient woke up on the next day morning. The ABPM measurements were analyzed according to the cut-off values in the 2018 European Society of Hypertension and the European Society of Cardiology (ESH/ESC) guidelines [22].

### 2.4. Statistical Analysis of the Results

Statistical analysis was conducted using the statistical software Dell Statistica Version 13 (Dell Inc., Round Rock, TX, USA). For quantitative variables arithmetic means, standard deviations (SD), and range of values (Min and Max values) were calculated for the estimated parameters in the studied groups. The distribution of variables was tested using the W–Shapiro–Wilk test. Following, in the comparative analyses of the mean values of quantitative variables, in cases of quantitative variables with normal distribution a one-way parametric ANOVA test was used. In cases of the absence of a normal distribution of variables, a one-way, non-parametric Kruskal–Wallis ANOVA was used to test the null hypotheses. First, the ranks of Kruskal–Wallis were analyzed. Then, in the case of statistical significance in the rank analysis, the differences between individual pairs of mean values were tested using the z-test for multiple comparisons. The study group consisted of patients with suspected bruxism, therefore the size of the fraction in the sample size calculator is 15% (the estimated frequency of bruxism in the population), then subgroups were separated taking into account the severity of bruxism—and compared in ANOVA. The calculator that was used was the online sample size calculator (https://www.naukowiec.org/dobor.html, accessed on 12 August 2020).

According to that, an a priori sample size calculation was performed. Statistically significant differences between arithmetic means were identified using the least significant difference (LSD) post-hoc test. In order to detect an association between studied variables analysis of correlation was performed. In cases of quantitative variables with normal distribution, Pearson’s correlation coefficients r were estimated, whereas in cases of quantitative variables without normal distribution Spearman’s r coefficients were calculated. The results at the level of 2-sided *p* < 0.05 were accepted as statistically significant.

## 3. Results

The mean age of all subjects was 33.9 ± 11.2 years. The mean BMI of the studied group was 22.5 ± 3.9, which was within the normal range. The participants (*n* = 51) were classified into the following three groups: mild bruxism group (2 ≤ BEI < 4, *n* = 19), severe bruxism group (BEI ≥ 4, *n* = 24), and control group without bruxism (BEI < 2, *n* = 8). The mean polysomnographic parameters of the study group are presented in Table 1.

A comparison of the biochemical parameters of the patients demonstrated the statistically significant differences in plasma sodium concentrations among the studied groups (Figure 1). Whereas other analyzed electrolytes’ concentrations (Mg^2+^ (mg/dL), K^+^ (mmol/L), Ca^2+^ (mg/dL)) did not show statistically significant differences among considered groups, which are shown in Table 2. A decreased sodium concentration was found in the mild bruxism group (2 ≤ BEI < 4) compared to the controls (BEI < 2) as well as in the severe bruxism group (BEI ≥ 4) compared to controls (Figure 1).

The mean 24-h SBP was 109.86 ± 7.3 mm Hg, and the mean DBP was 68.26 ± 5.5 mm Hg in the entire study group. No statistically significant differences were observed between the various parameters of BP in mild (2 ≤ BEI < 4) and severe bruxers (BEI ≥ 4) compared to the controls. The various parameters of BP are presented in Table 3.

A statistically significant positive linear correlation was found between sodium concentration and daytime SBP variability (r = 0.32, *p* < 0.05) as well as between sodium concentration and nighttime DBP variability (r = 0.31, *p* < 0.05). The correlations between blood pressure parameters and the concentration of sodium are presented in Figure 2.

## 4. Discussion

The most important finding of this study was the decreased sodium concentration in sleep bruxers compared to nonbruxers. Moreover, a relationship between sodium level and BP variability parameters was found in the SB groups. SB is no longer considered a simple dental behavior of peripheral etiology but rather a complex neurological phenomenon of neurodegenerative and inflammatory etiology with cardiovascular consequences [23]. Data concerning the association between sodium levels and sleep disorders are scarce. It was demonstrated that a lower serum level of sodium may be associated with insomnia [6]. Furthermore, sympathetic nerve activity due to insomnia [24,25] may lead to increased renin release and tubular fluid reabsorption [26] and may also be associated with low plasma sodium. Besides, this study analyzed the sodium level in patients with SB. To our best knowledge, for the first time, this study revealed decreased sodium concentration in sleep bruxers compared to healthy controls. As mentioned earlier, increased sympathetic nerve activity [27] is also seen among bruxers [23], which would explain a higher likelihood of lower serum sodium concentration among patients with SB as was observed in this study. In addition, a decreased level of sodium may indicate the fact that sodium is regulated by the antidiuretic hormone [28]. The antidiuretic hormone influences renal water reabsorption, by decreasing the excretion of free water to dilute plasma and lowering serum osmolality. Thus, lower serum osmolality manifests the lower sodium level. It is well known that the renal actions of the antidiuretic hormone are associated with water metabolism rather than directly with sodium itself. In conclusion, changes in the excretion of the antidiuretic hormone and sodium level affect BP in patients with SB [29]. However, this hypothesis needs further research.

Moreover, serotonin receptors have an effect on BP along with playing a role in regulating the serum level of sodium [30]. As stated in many previous studies, serotonin greatly influences BP through the receptors located on the most important regulating organs involved in cardiovascular regulation. This explains the effect of sympathetic nerve activity on cardiovascular tissues [31,32]. It was also shown that medications that increase the levels of serotonin also increase the serum level of sodium, for example, among the patients treated for depression [33]. Moreover, studies show that hyponatremia could be associated with the use of selective serotonin reuptake inhibitors (SSRI) medication as well as with serotonin syndrome [34]. It is worth noting that the relationship between the serotonin pathway and SB has been previously confirmed [35,36]. This suggests the relationship between the serotonin pathway and sodium concentration as well as between serum serotonin concentration and the severity of bruxism. The results of the present study are in agreement with previous research. Increased variability of BP observed in patients with SB could also be a risk factor for hypertension, influenced by plasma sodium concentration. This may indicate the factors affecting BP among patients with SB. Further research is needed to explain the pathomechanisms behind lower sodium plasma among bruxers, especially the pathophysiology of bruxism in arterial hypertension patients with an analysis of biochemical morphology parameters.

This study did not show statistically significant differences in the concentrations of magnesium, calcium, and potassium in bruxers compared to the controls. The levels of electrolytes may be associated with the intensity of bruxism, and magnesium treatment has also been attempted [37]; however, there is no evidence for the effectiveness of such a procedure. Although many studies have demonstrated an association between SB and vitamin D deficiency and low dietary calcium intake [38], researchers proved that the development of SB could be associated with increased scores of anxiety and depression (also observed in connection with the low serum level of vitamin D). Adding more, it was reported that vitamin D takes part in immune regulation, and its deficiency can alter immunomodulation and enhance the production of cytokines that play a role in sleep [39]. In this study, the levels of magnesium, potassium, and calcium were similar in nonbruxers and in the groups with bruxism, although more research may be needed on this topic.

The main strength of this study was the use of polysomnography, which is a gold standard in the diagnosis of SB. Moreover, the examination was conducted in controlled conditions of a sleep laboratory with video recording, which increases the accuracy of the SB diagnosis. The study also has the following limitations: plasma osmolality and the level of the antidiuretic hormone were not taken into consideration, and the size of the subgroup of nonbruxers was not large. The report is a preliminary report and definitely, a future examination in that field is needed. What is more, almost all Na- concentration was within the reference range of 135–145 mmol/L, and that would be very interesting to see whether there is a cut-off point there that the risk of SB increases with a lower concentration of plasma sodium.

## 5. Conclusions

1.The plasma concentration of sodium was decreased in bruxers compared to nonbruxers.2.A relationship exists between plasma sodium concentration and BP variability among patients with SB.3.As the results are present in a preliminary study, there is a need for further studies on the pathomechanisms of SB and their relationship with electrolyte plasma concentration as well as the inflammation factors.

## Figures and Tables

**Figure 1 biomedicines-10-02804-f001:**
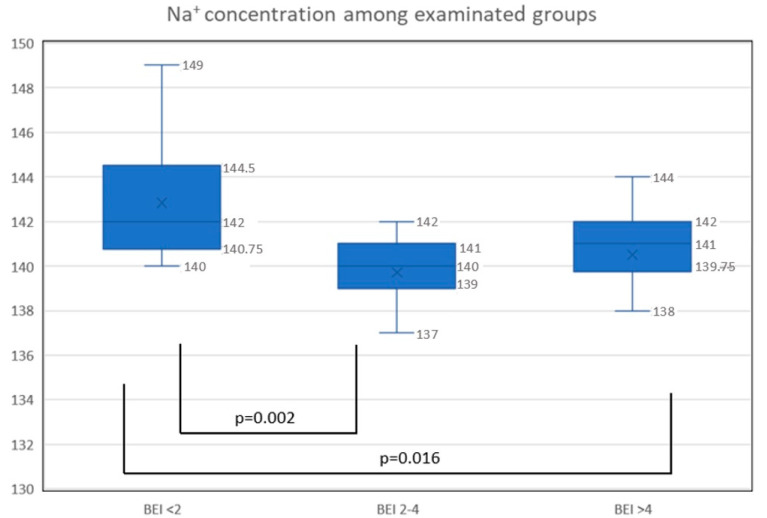
The concentration of sodium in examinated groups: nonbruxers (BEI < 2), mild bruxers (2 ≤ BEI < 4), and severe bruxers (BEI ≥ 4); Na^+^, sodium (mmol/L); *p*-value < 0.05 statistically significant.

**Figure 2 biomedicines-10-02804-f002:**
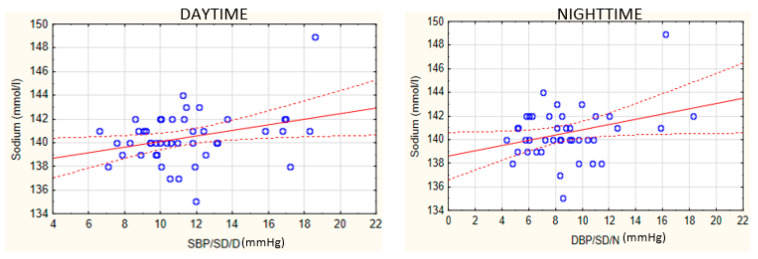
Correlations between BP parameters and the concentration of sodium. SBP, systolic blood pressure; DBP, diastolic blood pressure; SD, standard deviation; D, daytime; N, nighttime. Blue dots represent the data. The red dotted lines represent the confidence interval and the red line stands for the linear trend.

**Table 1 biomedicines-10-02804-t001:** Polysomnographic characteristics of the study group (*n* = 51).

	Mean	Minimum	Maximum	SD
AHI (n/h)	5.21	0.00	41.20	9.41
ODI (n/h)	4.68	0.00	39.10	8.01
BEI (n/h)	4.94	0.40	12.30	3.54
Snore (%)	6.68	0.00	68.8	14.89
TST (min)	418.3	153.50	523.50	68.77
SL (min)	23.89	1.70	163.50	24.06
WASO (min)	33.47	3.00	206.50	34.16
N1 (%TST)	5.10	0.30	17.90	3.98
N2 (%TST)	50.31	34.00	73.10	8.16
N3 (%TST)	21.90	6.90	39.40	7.19
REM (%TST)	22.69	6.30	35.20	5.28
ArI (n/h)	5.71	0.00	21.90	4.10
Mean SpO_2_ (%)	94.80	85.30	96.50	1.84
Minimum SpO_2_ (%)	88.88	57.00	96.00	6.53
SpO_2_ < 90% (%)	0.87	0.00	13.40	2.63

n/h, number/hour; AHI, apnea–hypopnea index; ODI, oxygen desaturation index; BEI, bruxism episode index; TST, total sleep time (min); SL, sleep latency; WASO, wake after sleep onset; N1, sleep stage 1; N2, sleep stage 2; N3, sleep stage 3; REM, rapid eye movement sleep stage; ArI, arousal index; mean SpO_2_, mean oxygen saturation; minimum SpO_2_, minimal oxygen saturation; SpO_2_ < 90%, mean oxygen saturation below 90%; SD, standard deviation.

**Table 2 biomedicines-10-02804-t002:** Concentrations of electrolytes in the entire study group and in nonbruxers (BEI < 2), mild bruxers (2 ≤ BEI < 4), and severe bruxers (BEI ≥ 4).

	2 ≤ BEI < 4	BEI ≥ 4	BEI < 2	Entire Group	*p*-Value
Mg^2+^ (mg/dL)	0.89 ± 1.8	1.39 ± 2.1	1.48 ± 2.3	1.25 ± 2.0	0.801
K^+^ (mmol/L)	4.3 ± 0.6	4.2 ± 0.3	4.6 ± 0.3	4.3 ± 0.4	0.445
Ca^2+^ (mg/dL)	9.1	9.2	9.3	9.2 ± 0.2	0.539

Mg^2+^, magnesium; K^+^, potassium; Ca^2+^, calcium; BEI, bruxism episode index.

**Table 3 biomedicines-10-02804-t003:** Ambulatory BP characteristics of the entire study group and the subgroups.

	Variable		Entire Group	BEI < 2	2 ≤ BEI < 4	BEI ≥ 4
SBP (mm Hg)	24-h mean	Average	109.8 ± 7.3	112.6 ± 7.1	110.1 ± 7.4	108.9 ± 7.4
		Variability	12.5 ± 3.1	12.7 ± 4.3	12.5 ± 2.9	12.6 ± 2.8
		Minimum	84.6 ± 7.0	89.1 ± 8.2	85.6 ± 6.4	82.4 ± 6.6
		Maximum	141.9 ± 14.5	140.0 ± 5.7	146.7 ± 17.3	138.7 ± 13.3
		Decline (%)	10.1 ± 6.3	7.1 ± 6.2	11.1 ± 8.1	10.1 ± 4.6
	Daytime	Average	113.0 ± 8.0	115.0 ± 7.2	113.7 ± 8.60	111.9 ± 8.0
		Variability	11.3 ± 2.8	10.5 ± 3.5	11.4 ± 2.8	11.4 ± 2.9
		Minimum	89.8 ± 8.2	91.4 ± 8.2	90.6 ± 8.5	88.6 ± 8.1
		Maximum	141.8 ± 14.5	140.0 ± 5.7	146.7 ± 17.4	138.3 ± 13.0
	Nighttime	Average	101.3 ± 8.1	106.3 ± 8,7	100.8 ± 7.3	100.2 ± 8.1
		Variability	9.4 ± 3.2	9.7 ± 3.1	7.7 ± 2.6	8.1 ± 10.6
		Minimum	85.4 ± 7.9	91.1 ± 9.2	87.7 ± 8.0	81.8 ± 5.7
		Maximum	120.8 ± 12.2	128.8 ± 8.3	115.8 ± 10.3	122.0 ± 12.3
DBP (mm Hg)	24-h mean	Average	68.2 ± 5.6	70.3 ± 7.5	67.3 ± 4.9	68.4 ± 5.5
		Variability	11.3 ± 2.8	10.7 ± 3.2	11.7 ± 2.8	11.1 ± 2.8
		Minimum	46.4 ± 5.7	48.4 ± 8.4	46.5 ± 5.4	45.7 ± 5.1
		Maximum	100.2 ± 16.8	98.7 ± 11.8	105.8 ± 18.9	96.3 ± 15.6
		Decline (%)	15.4 ± 7.8	11.8 ± 9.4	18.8 ± 16.7	15.3 ± 4.6
	Daytime	Average	71.3 ± 6.1	72.8 ± 7.4	70.5 ± 6.3	71.4 ± 5.7
		Variability	10.3 ± 3.1	9.7 ± 3.4	10.8 ± 3.1	10.1 ± 3.1
		Minimum	50.8 ± 7.1	52.4 ± 6.8	50.7 ± 6.9	50.5 ± 7.5
		Maximum	99.5 ± 16.4	98.5 ± 11.9	105.8 ± 18.8	94.9 ± 14.2
	Nighttime	Average	60.1 ± 6.4	63.8 ± 9.9	58.6 ± 5.5	60.1 ± 5.6
		Variability	8.5 ± 3.0	9.2 ± 3.4	7.4 ± 2.8	9.1 ± 2.8
		Minimum	46.8 ± 5.5	48.5 ± 8.7	47.4 ± 5.8	45.7 ± 3.7
		Maximum	78.4 ± 13.0	82.8 ± 11.1	75.1 ± 12.6	79.5 ± 13.0
MAP (mm Hg)	24-h mean	Average	82.6 ± 5.6	84.7 ± 7.0	82.2 ± 4.9	82.3 ± 5.7
		Variability	10.8 ± 2.5	10.5 ± 2.9	11.1 ± 2.5	10.7 ± 2.5
		Minimum	61.3 ± 5.6	62.3 ± 8.1	61.7 ± 5.5	60.7 ± 5.2
		Maximum	113.9 ± 14.7	113.6 ± 9.3	118.5 ± 16.9	110.4 ± 13.8
		Decline (%)	12.1 ± 6.4	10.0 ± 7.2	13.2 ± 8.4	11.9 ± 4.2
	Daytime	Average	85.4 ± 6.2	87.0 ± 6.9	85.2 ± 6.3	85.1 ± 6.1
		Variability	9.9 ± 2.7	9.4 ± 3.2	10.2 ± 2.5	9.7 ± 2.7
		Minimum	65.4 ± 6.4	66.7 ± 6.9	65.4 ± 7.0	65.0 ± 5.9
		Maximum	112.1 ± 8.7	118.5 ± 16.9	109.0 ± 12.8	113.0 ± 14.5
	Nighttime	Average	75.0 ± 9.1	78.3 ± 9.1	74.1 ± 5.5	74.7 ± 6.1
		Variability	8.0 ± 3.0	9.2 ± 3.3	6.7 ± 2.5	8.7 ± 3.0
		Minimum	61.8 ± 6.0	62.7 ± 8.8	63.5 ± 6.3	60.2 ± 4.4
		Maximum	92.2 ± 13.1	98.7 ± 14.1	88.5 ± 11.4	92.9 ± 13.6

SBP, systolic blood pressure; DBP, diastolic blood pressure; MAP, mean arterial pressure; BEI, bruxism episode index.

## Data Availability

The data presented in this study are available on reasonable request from the corresponding author.

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
