# Peer review of "The Relationship between the Plasma Concentration of Electrolytes and Intensity of Sleep Bruxism and Blood Pressure Variability among Sleep Bruxers"

_biomedicines, 2022, doi:10.3390/biomedicines10112804_

Round 1

Reviewer 1 Report

I read with interest this brief study concerning plasma sodium levels in subject affected by sleep bruxism.  Although interesting, this study owns some serious lacks, especially in material and methods, that can be obviously solved by the authors. In addition,  I found the results very preliminary and not  supportive of the conclusions stated by the authors. Some explanations might help to improve the paper, but concerns still remains for the main issue. 

Below the authors can find the comments concerning the main issue and some minor ones.

MAIN ISSUE:

I have a big concern: Why only 8 subjects were included in the control group (without SB)? Based on this small sample I think that the findings are not conclusive.  The three compared groups are in fact unequally in size and can lead to several problems (loss of statistical power, unequal variance and cofounding variables). 

Given the large disparity in the sample sizes, I would suggest an unequal variances version of the t-test and at least a detailed discussion on this crucial point. But I found no mention on which statistical test has been used for this verification. 

By reading the entire paper I can not find strong results supporting  the conclusions stated by the authors: both, discussion and conclusion should have focused also on the main above mentioned issue/limitation of this study, and more caution should be paid given that that this study has numbers supporting only very preliminary results.

MINOR:

Row 54: Please explain better the concept of interdisciplinary behaviour or re-wording the phrase.

Row 59: This statement needs a reference. 

Row 75: SB was diagnosed (ascertained) or just assumed? Please give details on this aspect and eventually add a brief explanation on the diagnosis of SB in the introduction section.

Row 79: Is body mass index in the normal range? Please specify it.

·      Please describe in the material and methods section the three groups analysed: controls, severe and mild SB.

·      Please describe what statistical tests have been used for verifying he differences between the groups. 

Author Response

Dear Reviewer,

On behalf of our research team, I would like to attach our response below. I would like to thank you for your review of our article and your suggestions which helped us to improve our manuscript. Please see the attachement.

Kind regards,

Justyna Kanclerska

Reviewer 2 Report

All my comments and suggestions are included in the pdf file!

Author Response

Dear Reviewer,

On behalf of our research team, I would like to attach our response below. I would like to thank you for your review of our article and your suggestions which helped us to improve our manuscript. Please see the attachment.

Kind regards, 

Justyna Kanclerska

Round 2

Reviewer 1 Report

1) There is still the lack of explanation of the three sample groups in M&M section, including the control group. Please add this piece of information that suddenly appear only at the result section. 

2)something is wrong with row 146: is a caption or a text period? Anyway why table 33?

3)Please revise the sentence in row 157:

'A comparison of the biochemical parameters of the patients demonstrated the statistically significant differences in plasma sodium concentrations among the studied groups  (Table 2; Figure 1)'

In fact, in table 2 are reported other plasma electrolyte levels (which are not significantly different among the groups), while sodium levels are missing from this table and only depicted in figure 2. Please solve this mess.

4) Please add 'preliminary' before results in row 32 (abstract)

5) I still do not like the conclusions, which should be more cautious (I am specifically referring to the sentence in row 249). Please specify that the conclusion reflects only preliminary results. 

A suggestion for future revisions: please highlight or colore differently all the modifications made to the original paper in the revision phase in order to facilitate the reviewer work!

Author Response

Thank you for the review, please see the attachment file with the response.

Reviewer 2 Report

Thanks for your effort to consider my suggestions! However, I still have some minor remarks:

 Please correct formulation of the following sentences:

(a)    ‘In cases of variables manifesting distribution distinct than the normal one the nonparametric equivalent of analysis of variance, i.e. ANOVA test of Kruskal-Wallis was used.’

 I’m not quite sure what you mean with ‘NIR’ test? Maybe this sentence needs reformulation as well?

(b)    ‘Statistically significant differences between arithmetic means were identified using the post-hoc NIR test.’

(c)     ‘In order to detect relationships between studied variables analysis of correlation was performed. In cases of quantitative variables manifesting normal distribution, Pearson’s correlation coefficients r were estimated, in cases of quantitative variables manifesting distribution distinct than the normal one Spearman’s r coefficients were calculated.’

All the information regarding sample size calculation is missing or at least I am unable to find it??? Line 146 is quite confusing…. Can you point to the line were you mentioned

(a) the calculator used,

(b) apriori sample size calculation,

(c) a clearly stated primary endpoint is still missing, isn’t it?!,

(d) we estimated frequency of bruxism in the Polish population

I’m still confused with your explanation regarding correction (‘In cases of variables manifesting distribution distinct than the normal one the nonparametric equivalent of analysis of variance, i.e. ANOVA test of Kruskal-Wallis was used. Statistically significant differences between arithmetic means were identified using the post-hoc NIR test. One of the post-hoc tests was used, the NIR test was subjectively chosen.’). Distribution and correction for multiple testing are separate topics-please clarify!

Author Response

Thank you for the review. We included the response in the attached file. Please see the attachment below.

Round 3
